# Three-Fluorophore FRET Enables the Analysis of Ternary Protein Association in Living Plant Cells

**DOI:** 10.3390/plants11192630

**Published:** 2022-10-06

**Authors:** Nina Glöckner, Sven zur Oven-Krockhaus, Leander Rohr, Frank Wackenhut, Moritz Burmeister, Friederike Wanke, Eleonore Holzwart, Alfred J. Meixner, Sebastian Wolf, Klaus Harter

**Affiliations:** 1Center for Plant Molecular Biology (ZMBP), University of Tübingen, 72076 Tübingen, Germany; 2Institute for Physical & Theoretical Chemistry, University of Tübingen, 72076 Tübingen, Germany; 3Centre for Organismal Studies (COS), University of Heidelberg, 69117 Heidelberg, Germany

**Keywords:** protein-protein interaction, plasma membrane, nanodomains, spectral Förster resonance energy transfer (FRET), FRET-fluorescence lifetime imaging microscopy (FRET-FLIM)

## Abstract

Protein-protein interaction studies provide valuable insights into cellular signaling. Brassinosteroid (BR) signaling is initiated by the hormone-binding receptor Brassinosteroid Insensitive 1 (BRI1) and its co-receptor BRI1 Associated Kinase 1 (BAK1). BRI1 and BAK1 were shown to interact independently with the Receptor-Like Protein 44 (RLP44), which is implicated in BRI1/BAK1-dependent cell wall integrity perception. To demonstrate the proposed complex formation of BRI1, BAK1 and RLP44, we established three-fluorophore intensity-based spectral Förster resonance energy transfer (FRET) and FRET-fluorescence lifetime imaging microscopy (FLIM) for living plant cells. Our evidence indicates that RLP44, BRI1 and BAK1 form a ternary complex in a distinct plasma membrane nanodomain. In contrast, although the immune receptor Flagellin Sensing 2 (FLS2) also forms a heteromer with BAK1, the FLS2/BAK1 complexes are localized to other nanodomains. In conclusion, both three-fluorophore FRET approaches provide a feasible basis for studying the in vivo interaction and sub-compartmentalization of proteins in great detail.

## 1. Introduction

Integration of different signaling cues at the cellular level is essential for the survival of any organism. In plants, for instance, mechanical damage to the cell wall causes attenuation of cellular growth response while resources are redistributed to repair processes. With the advent of high- and super-resolution microscopy techniques, the discovery of protein localization in nanodomains and the spatial organization of receptors and accompanying proteins (e.g., co-receptors) in the plasma membrane (PM) have come into focus. This has prompted new questions regarding the extent to which the constituents of a given signaling complex or nanodomain are integrated or are disintegrated upon signal perception.

A well-understood pathway in plants is the brassinosteroid (BR) hormone signal transduction, which is mediated by the PM-resident receptor kinase Brassinosteroid Insensitive 1 (BRI1) and its co-receptor BRI1 Associated Kinase 1 (BAK1). The binding of BR to BRI1’s extracellular domain increases its interaction with BAK1 and a re-arrangement of proteins in the complex, eventually leading to the auto- and trans-phosphorylation of their Ser/Thr-kinase domains [1,2]. These PM-resident events result in the differential regulation of BR-responsive genes through a nucleo-cytoplasmic signaling cascade [3,4,5,6] and the activation of PM-resident P-type proton pumps [7,8].

Receptor-Like Protein 44 (RLP44) was previously described to play a role in cell wall integrity sensing via modulation of BRI1/BAK1-dependent signaling [9] as well as to interact with BRI1 and BAK1 [9,10]. This suggests that RLP44 directly affects the activity of the BRI1/BAK1 complex upon input from the cell wall, as a scaffold protein for the establishment of a specific BRI1/BAK1 complex in the PM [2].

To test the hypothesis of ternary RLP44/BRI1/BAK1 complex formation, we first wanted to establish a three-fluorophore, intensity-based spectral Förster resonance energy transfer (FRET) as well as -fluorescence lifetime imaging microscopy (FRET-FLIM) technique in plant cells. FRET is the non-radiative energy transfer from a donor to an acceptor fluorophore by dipole-dipole interaction, which is only possible over small distances and depends on the relative dipole moment orientation of the donor and acceptor fluorophores [11]. FRET manifests itself by an alteration of the donor and acceptor fluorescence intensities, but also decreases the donor´s excited state lifetime due to the additional relaxation path from the donor to the acceptor [12]. In contrast to two-fluorophore FRET, an intermediate acceptor fluorophore is included in the energy transfer pathway of three-fluorophore FRET. As the plant cell wall cannot be penetrated by organic dyes, our approach depends solely on genetically encoded fluorophores. Here, we exemplary choose monomeric Turquoise 2 (mTRQ2) as donor, monomeric Venus (mVEN) as the first acceptor and monomeric red fluorescence protein 1 (mRFP) as the second acceptor.

Based on theoretical calculations we experimentally show by intensity-based spectral FRET and FRET-FLIM that RLP44, BRI1 and BAK1 are unified in a ternary complex that likely forms in a distinct nanodomain in the PM. This nanodomain is spatially clearly distinct from the FLS2/BAK1 complex-containing nanodomain. In addition, we propose FRET-FLIM to be always better than intensity-based FRET for the analysis of ternary protein complex formation and relative proximity estimates in plant cells.

## 2. Results

### 2.1. Physicochemical Properties of the Used Fluorophores

As the selected fluorophores substantially influence the quality of the FRET data, we screened a variety of fluorophores for their suitability in our approach. As a result, all of the used genetically encoded fluorophores were monomeric, minimizing false-positive FRET originating from aggregation. mTRQ2 was used as donor. It has numerous advantages including a long, mono-exponential fluorescence lifetime (FLT) [13] and the spectral overlap of mTRQ2 emission with the absorbance of the first acceptor mVEN is high, yielding a large Förster distance R_0_ [14,15,16]. The second acceptor is mRFP, which has a large spectral overlap with the first acceptor/second donor mVEN. The maturation time of the fluorophores is also a crucial factor for the FRET efficiency (E_FRET_) [17,18]. In our three-fluorophore system, both mVEN and mRFP have faster maturation times than mTRQ2, ensuring favorable FRET conditions (more details about the fluorophores in Appendix A).

### 2.2. Calculated Properties of the mTRQ2/mVEN/mRFP Three-Fluorophore FRET System

To determine the operational FRET range for the chosen fluorophores, the corresponding Förster distances (R_0_) were calculated (Table 1) as well as the distance corresponding to 10% FRET efficiency (r_10%_). This last parameter best illustrates the measurement limit for FRET that can be realized with standard FRET microscopy measurements. For the complex energy flow in a three-fluorophore setting, r_10%_ was calculated for the mTRQ2/mRFP pair, adding mVEN as intermediate acceptor. According to the absorption and emission spectra of the used fluorophores (Figure 1a), the emission of mTRQ2 shows a significant spectral overlap with the absorption spectra of both mVEN and mRFP (Figure 1a). Furthermore, the emission spectrum of mVEN substantially overlaps with the absorption spectrum of mRFP, enabling energy transfer from mTRQ2 to mVEN, mVEN to mRFP and mTRQ2 to mRFP. R_0_ was calculated with n = 1.4 and κ^2^ = 2/3 according to [19], using the spectra and photophysical data from the original publications (Appendix A). The resulting Förster distances R_0_ were 5.7 nm for the mTRQ2/mVEN, 5.2 nm for the mVEN/mRFP and 5.1 nm for the mTRQ2/mRFP pair (Table 1, Figure 1b). Evidently, the same trend is observed for r_10%_, which is highest for the mTRQ2/mVEN pair and lowest for the mTRQ2/mRFP pair (Table 1, Figure 1b). In addition, FRET was calculated not to be limited to sequential energy transfer from mTRQ2 via mVEN to mRFP but also from mTRQ2 to mRFP directly (Table 1, Figure 1c).

For large distances between mTRQ2 and mRFP no direct FRET is expected to be possible. Therefore, the introduction of the intermediate acceptor mVEN could increase the dynamic range between mTRQ2 and mRFP. To estimate this long-range effect, we investigated how the presence of mVEN affects the energy transfer to mRFP in such a FRET cascade [21] and calculated an increase for r_10%_ from 8.2 nm to 12.4 nm (Figure 1c, Table 1), which is in agreement with the increase in effective FRET distance reported previously [22]. However, the authors of [22] fixed the fluorophore positions along a DNA helix with equidistant separation between the fluorophores. In multimeric protein complexes, however, neither the complex geometry nor the exact position of the attached fluorophores are usually known. Therefore, we additionally calculated an averaged r_10%_ value when mVEN is inserted at random positions between mTRQ2 and mRFP. In this more realistic representation of our dynamic plant system, we found a slightly smaller increase of r_10%_ from 8.2 nm to 11.1 nm (Figure 1c, Table 1). This is by far large enough to be able to detect long-range interactions with our three-fluorophore FRET approach.

### 2.3. Structural Simulation of the Arrangement of the Fluorophore-Tagged Proteins for the Estimation of the FRET Range

To assess how the maximum dynamic range of about 11.1 nm relates to the size of our proteins of interest, we arranged the cytoplasmic domains of RLP44, BRI1 and BAK1 fused to the respective fluorophores and the calculated r_10%_ values between mTRQ2, mVEN and mRFP in scale in a graph (Figure 1d). As the structures of the fluorophores and the cytoplasmic domains of BRI1 and BAK1 are available, a solvent-accessible surface representation was generated to estimate the protein sizes [23,24,25]. The orientation of the BRI1 and BAK1 kinase domains to each other was depicted according to the highest probability after a molecular docking analysis [20]. The structures of RLP44´s cytoplasmic domain and the Gateway^®^-linkers were not available and predicted with PEP-FOLD3 [26] (for details see Section 4). The stoichiometry-adjusted r_10%_ values for all FRET pairs are between 7.3 and 8.2 nm (Table 1) and therefore could span the distance of two kinase domains, which have diameters of about 4.8 nm (Figure 1d). Most importantly, while RLP44-mTRQ2 and BAK1-mRFP are too far apart for FRET in this exemplary linear arrangement, three-fluorophore FRET from RLP44-mTRQ2 via BRI1-mVEN to BAK1-mRFP is able to span this distance (Figure 1d). The same is expected for a RLP44-mTRQ2/BAK1-mVEN/BRI1-mRFP arrangement.

### 2.4. Calculation of Cross-Excitation and Bleed-Through in Intensity-Based Spectral FRET Measurements

The vast majority of investigations in animal cells have assessed three-fluorophore FRET with intensity-based methods, specifically by quantitative acquisition of spectra, using predominantly organic dyes. Therefore, we decided to first assess the ternary complex formation for the selected fusion proteins in plant cells with this method.

Regardless of the stoichiometry of the complexes, the expression levels of the fluorophore-fused proteins have a large influence on the spectra of intensity-based FRET, due to cross-excitation and bleed-through. This is demonstrated in the simulated emission spectra of Figure 2, calculated for a complex stoichiometry of 1:1:1 and a linear arrangement of mTRQ2, mVEN and mRFP placing mVEN at equal distance to the other two fluorophores. With this fluorophore ratio, inter-fluorophore distances above 10 nm (no FRET) and an excitation wavelength of 458 nm, an intensity peak at around 525 nm was calculated to appear due to cross-excitation (Figure 2a, top). With inter-fluorophore distances of 7 nm (Figure 2a, middle) and 5 nm (Figure 2a, bottom), the energy transfer from mTRQ2 to mVEN and mRFP became apparent in the simulated spectra, as the relative mTRQ2 intensity peak decreased, while the mVEN and mRFP peaks increased. The theoretical bleed-through and cross-excitation depend strongly on the donor-to-acceptor ratio of the fluorophores (Appendix A). Importantly, the bleed-through and cross-excitation effects have a larger influence on the apparent mVEN signal than FRET itself: Already a ratio of 1:2 for mTRQ2/mVEN results in a peak of the fluorescence intensity at around 525 nm similar to that calculated for an inter-fluorophore distance of 7 nm (compare Figure 2a, middle and Figure 2b, top). However, the different mRFP intensity progression was well observable (compare Figure 2a with Figure 2b). This underscores the necessity for a careful calibration of the fusion protein amounts. Therefore, the bleed-through and cross-excitation are always quantified in the presented absorption and emission spectra as well as in the confocal images (see Appendix A).

### 2.5. Experimental Determination of Cross-Excitation and Bleed-Through in Plant Cells

It was previously shown that BRI1/BAK1, RLP44/BRI1 and RLP44/BAK1 form heteromers in plant cells [9,10,27]. In contrast, the PM-localized immune response mediating Flagellin Sensing 2 (FLS2) receptor does not interact with RLP44 [28,29] and was selected as a negative control in the further analyses. The spectra were acquired in transiently transformed *Nicotiana benthamiana* epidermal leaf cells. All fusion proteins localized to the PM (Appendix A). The procedure of spectra acquisition from the regions of interest in the PM is described in detail in Appendix A.

The fluorescence emission spectra of the three fusion constructs (RLP44-mTRQ2, FLS2-mVEN and BAK1-mRFP) were analyzed separately under excitation at 458 nm, whereby only FLS2-mVEN showed a significant amount of cross-excitation (Figure 3a). The quantification of the fluorescence emission of RLP44-mTRQ2 after excitation at 458 nm is a measure of its protein level, manifesting an average intensity of 120 arbitrary units (a.u.) (Figure 3a,b). It was reflected in an emission peak intensity of 120 a.u. in the spectrum (Figure 3a). A relative high accumulation level of FLS2-mVEN with an average intensity of 200 a.u. after excitation at 514 nm (Figure 3b) caused a much smaller background signal of 8 a.u. in the spectrum after excitation with light of 458 nm (Figure 3a). The excitation of BAK1-mRFP with light of 561 nm gave an average intensity of 110 a.u. (Figure 3b) but did not result in a distinct peak in the emission spectrum after excitation with light of 458 nm (Figure 3a). Thus, the different accumulation levels of the fusion proteins led to the same shape of the fluorescence emission spectra but with variations in the peak intensities (Appendix A).

### 2.6. Intensity-Based FRET Analysis of Dual Protein-Protein Interactions

First, we assessed FRET from the mTRQ2 to the mVEN fusion proteins. Since our simulations indicate that the amount of the fusion proteins has a larger influence on the shape of the spectra than FRET itself, a correction for the relative proteins levels was imperative. Therefore, it was always necessary to compare the spectra with the same donor-to-acceptor ratios of RLP44-mTRQ2/BRI1-mVEN with that of RLP44-mTRQ2/FLS2-mVEN. To this end, each recorded spectrum was subjected to spectral unmixing to determine the relative proportions of mTRQ2 and mVEN (Appendix A). Then, the spectral unmixing information was combined with the respective estimates of fusion protein levels. To do so, each channel was imaged with sequential excitation before acquisition of the spectra (excitation with only 458 nm, then only 514 nm, lastly only 561 nm).

In the spectra with an identical donor-to-acceptor sample ratio, the peak intensity value at about 525 nm was higher for the RLP44-mTRQ2/BRI1-mVEN than for the RLP44-mTRQ2/FLS2-mVEN sample (Figure 3c). The integration of the spectral unmixing results and donor-to-acceptor ratios revealed that the proportion of mVEN emission in the spectra of the RLP44-mTRQ2/BRI1-mVEN sample differed significantly from that of the RLP44-mTRQ2/FLS2-mVEN sample (Figure 3d). This was also true for donor-to-acceptor ratios of approximately 1:10 and 1:4 (Figure 3d). In conclusion, FRET from RLP44-mTRQ2 to BRI1-mVEN was observed when compared with the non-FRET RLP44-mTRQ2/FLS2-mVEN control pair.

We next tested, whether an energy transfer from the mTRQ2 to mRFP fusion proteins was at all possible. No significant increase of the emission at around 610 nm was detected compared to the donor alone control for the co-accumulation of RLP44-mTRQ2 with FLS2-mRFP, (Figure 3e). In contrast, co-accumulation of RLP44-mTRQ2 with BAK1-mRFP or BRI1-mRFP led to a significant increase in the emission at around 610 nm compared to the control (Figure 3e). The fluorescence intensity ratio for the mTRQ2 and mRFP was 1:1 for the RLP44-mTRQ2/FLS2-mRFP pair (Figure 3f), whereas for RLP44-mTRQ2/BRI1-mRFP and RLP44-mTRQ2/BAK1-mRFP it was 1:2 and 1:5 respectively, i.e., higher than in the RLP44-mTRQ2/FLS2-mRFP control (Figure 3f). After correction for the different fusion protein amounts, spectrally detectable FRET clearly was observed from RLP44-mTRQ2 to BAK1-mRFP and to BRI1-mRFP but not to FLS2-mRFP.

### 2.7. Intensity-Based FRET Analysis of Ternary Protein Complex Formation

When RLP44, BRI1 and BAK1 form a ternary protein complex, the average distance between RLP44 and BAK1 might be altered by the presence of BRI1. For example, BRI1 may be located between RLP44 and BAK1, increasing the distance between them. Also the total number of RLP44/BAK1 pairs may be reduced, as additional pairs such as RLP44/BRI1 or BAK1/BRI1 can be formed [30].

To test this possibility, we expressed BRI1 with a non-fluorescent HA-tag (BRI1-HA) in tobacco cells (Appendix A). Co-expression of BRI1-HA did not change the intensity of the RLP44-mTRQ2 spectrum (compare Figure 4a and Figure 3a). As described above, co-accumulation of RLP44-mTRQ2 with BAK1-mRFP yielded a FRET-induced emission peak at around 610 nm (Figure 4a). This peak was no longer present when BRI1-HA was co-expressed with RLP44-mTRQ2 and BAK1-mRFP (Figure 4a). In contrast, the co-expression of RLP44-mTRQ2/FLS2-mVEN/BRI1-mRFP retained the intensity peak at around 610 nm (Appendix A). Thus, the addition of BRI1-HA increased the average distance between RLP44-mTRQ2 and BAK1-mRFP or led to alterations in the fluorophore-tagged protein levels. As the latter was not the case (Figure 4b), our results demonstrate that the distance between RLP44-mTRQ2 and BAK1-mRFP did indeed change upon BRI1-HA co-accumulation. When BRI1-mVEN was co-expressed instead of BRI1-HA, the emission peak at around 610 nm reappeared (Figure 4c,d). At the same time, energy transfer from mVEN to mRFP was detected, as the emission peak of mVEN at 525 nm strongly decreased in the RLP44-mTRQ2/BRI1-mVEN/BAK1-mRFP sample in comparison with the RLP44-mTRQ2/BRI1-mVEN sample which was not observed in the FLS2 control samples (Figure 4a,c). Although the influence of cross-excitation of mVEN and subsequent FRET directly to mRFP could not be excluded entirely, this effect is negligible, as no energy transfer from mVEN to mRFP was observed for the RLP44/FLS2/FLS2 sample (Appendix A). Likewise, the analysis of the different protein combinations did not change the donor-to-acceptor ratios significantly (Figure 4d). In conclusion, the appearance of the emission peak at around 610 nm, which was accompanied by the parallel reduction of the emission at around 525 nm, fulfils the criteria for three-chromophore FRET from RLP44-mTRQ2 via BRI1-mVEN to BAK1-mRFP. In corroboration, the energy transfer described above was also observed when the acceptor fluorophores were exchanged (Appendix A).

In conclusion, our data convincingly demonstrate that FRET from mTRQ2 via mVEN to mRFP specifically occurred for the RLP44/BRI1/BAK1 and RLP44/BAK1/BRI1 combinations, but not for the RLP44/BAK1/FLS2 combination. This indicates that RLP44, BRI1 and BAK1 form a specific ternary complex and/or are arranged in very close spatial proximity (5.7 to 13.7 nm) in the PM specifically. The successful establishment of intensity-based three-fluorophore FRET now prompted us to test whether the complex formation would also be reflected in FLIM measurements.

### 2.8. Measurement of In Vivo RLP44/BRI1/BAK1 Ternary Complex Formation by FRET-FLIM

When studying three-way interactions with intensity-based spectral methods, the concentration ratio between donor and acceptor molecules is of major importance, as spectral bleed-through and cross-excitation mimics potential FRET. In contrast, when monitoring the fluorescence lifetime (FLT) of the donor by FLIM, no such careful and labor-intensive calibrations are required [7,31]. We therefore tested, whether the complex formation and spatial arrangement of RLP44, BRI1 and BAK1 can also be monitored and confirmed by changes in the FLT of mTRQ2.

We determined an average FLT of about 3.99 ns for RLP44-mTRQ2 (Figure 5a) as reported before [13,15]. Interestingly, a significant decrease in the FLT is observed for the co-accumulation of RLP44-mTRQ2 with BRI1 fused to either mVEN (3.54 ns) or mRFP (3.60 ns) or with BAK1-mRFP (3.54 ns), indicating FRET from RLP44-mTRQ2 to BRI1-mVEN, BRI1-mRFP or BAK1-mRFP (Figure 5a). This constitutes the lower FLT limit that is expected when only heterodimers are present in the PM. Serving as a negative control, the co-expression of FLS2-mVEN (3.70 ns) or FLS2-mRFP (3.90 ns) with RLP44-mTRQ2 did not cause a significant decrease in mTRQ2´s FLT (Figure 5a,b). Furthermore, the co-accumulation of FLS-mRFP with RLP44-mTRQ2 and BRI1-mVEN did not further reduce the FLT of RLP44-mTRQ2 (3.60 ns) compared to the condition, where FLS2-mRFP is not present (Figure 5a). In contrast, the presence of BAK1-mRFP together with RLP44-mTRQ2 and BRI1-mVEN led to a significant further decrease in the FLT of RLP44-mTRQ2 to 3.40 ns, that is beyond the aforementioned values for the RLP44-mTRQ2/BRI1-mVEN heteromers of 3.54 ns (Figure 5a). This is only explainable by the existence of an additional FRET pathway that becomes possible if RLP44, BAK1 and BRI1 have formed a ternary complex. When 10 nM brassinolide (BL), an active brassinosteroid, was applied to the cells, no alteration in the FLT of RLP44-mTRQ2 was observed (Figure 5c). It was previously shown that FRET-FLIM is, in principle, independent of the donor concentration [32]. However, the relative amount of acceptor in comparison to the donor fluorophores (the donor-to-acceptor ratio) may influence the FLT of the donor [21,33]. Therefore, we investigated, whether this effect influences the FLT values obtained using different fluorescent protein ratios. As only the ratio of RLP44-mTRQ2/FLS2-mRFP and RLP44-mTRQ2/BAK1-mRFP exhibited a measurable difference within the RLP44-mTRQ2/BRI1-mVEN/FLS2-mRFP and the RLP44-mTRQ2/BRI1-mVEN/BAK1-mRFP in the three-fluorophore arrangements, differences in expression strength of the fusion proteins were not a major factor influencing the FLT of RLP44-mTRQ2 (Appendix A).

In summary, we were able to confirm the ternary complex formation of RLP44, BAK1 and BRI1 by three-fluorophore FRET-FLIM in living plant cells. Furthermore, the FRET-FLIM approach is clearly superior to the intensity-based FRET approach, because the former is easier to implement and less prone to errors.

## 3. Discussion

Many studies in the non-plant field have independently established fluorophore sets for studying binary protein-protein interactions and complex formation in the cellular context [34]. Here we present a three-fluorophore intensity-based FRET and a FRET-FLIM approach with the genetically encoded fluorophores mTRQ2, mVEN and mRFP in plant cells. The use of a blue, yellow and red fluorophore was the preferred application for three-fluorophore FRET studies, as it provides a good compromise between large Förster distances due to high spectral overlap and sufficient spectral separation for independent detection. A comparative study on available fluorophores in non-plant cells revealed that mTRQ2, YPet and mCherry is the most promising combination for three-fluorophore FRET [34]. However, YPet is not yet adapted for use in plant cells yet. In addition, even though mVEN is not as bright as YPet, it is a monomer [35], which is essential for FRET-based interaction studies. We reasoned that the slightly lower brightness of mVEN is less of a concern than the potential tendency of other fluorophores to form aggregates at high local concentrations in confined domains, which can be expected when they are fused to PM-resident proteins [35,36]. High brightness fluorophores, such as Ruby2 or TagRFPs, are outperformed by mCherry as it compensates its low brightness with its fast maturation rate [37]. The maturation time of mRFP, however, is also relatively short, which makes it an appropriate acceptor fluorophore as well. Until now, no study was able to circumvent direct FRET between the donor and the 2nd acceptor for genetically encoded fluorophores [21,37,38,39,40,41]. Therefore, extensive corrections for the crosstalk of the fluorophores are required in intensity-based FRET. While organic dyes might have provided better photophysical properties, their application in plant cells is challenging as the cell wall interferes with in vivo labelling. Even in animal cells, the different labelling efficiencies of organic dyes are challenging for three-fluorophore FRET, as the correction for directly excited acceptor fluorescence proves to be problematic [32,42,43].

As of yet, only one single study has tried to estimate the dynamic range of the distances between three fluorophores present in their set [22]. We think that a realistic estimation of the variability and restrictions of a three-fluorophore system is important for the interpretation of FRET data. For example, the general rule of thumb for reliable FRET measurements below 10 nm may be imprecise or incorrect. Here, we calculated adapted distance limits, taking several factors into account: First, the spatial arrangement of the fluorophores in a three-fluorophore setting and secondly, the possibility for multiple acceptors per donor for higher complex stoichiometries. Indeed, for an ideal linear arrangement of the fluorophores mTRQ2, mVEN and mRFP, FRET measurements can cover distances of up to 11.1 nm. This implies that FRET in membranes could not only occur between proteins within one and the same complex or domain, but might also be possible between proteins located in spatially distinct but adjacent domains (Figure 6).

When assuming that the primary donor is located between the 1st and 2nd acceptor in a ternary complex, the likelihood of energy transfer from the donor via acceptor 1 to the acceptor 2 decreases, as the distance between acceptors 1 and 2 increases. These spatial arrangements should be considered when applying intensity-based FRET measurements. Furthermore, in this context it is also recommended in this context to exchange at least one of the fluorophores between the fusion proteins, as it was done here. For classical donor-based FRET-FLIM, the differences in spatial organization may not be detectable, as no information is available on where the energy is transferred to. In principle, such spatial information can also be obtained using more sophisticated FRET-FLIM approaches, like when mVEN is used as donor by excitation with another laser. In any case, FLIM-based FRET measurements have the advantage of a higher sensitivity and require less processing, as the signal-to-noise ratio in intensity-based FRET measurements becomes worse after unmixing and sensitized emission calculations.

Since the PM is a compartment confined to two dimensions, the protein density is comparatively higher than in the three-dimensional case. In particular, membrane receptors tend to form complexes with higher stoichiometries, meaning that a donor can be surrounded by several acceptors, which can increase the apparent FRET efficiency. This can mask the principally linear relationship between E_FRET_ and the affinity properties of the interacting fusion proteins. Therefore, no conclusions can be drawn about their affinity within a ternary complex and only limited information can be obtained about the absolute distances between the three fluorophores.

It was previously unclear whether RLP44 interacts with both BRI1 and BAK1 simultaneously. In this study, we applied intensity-based FRET and FRET-FLIM to probe that RLP44 is specifically located in close proximity (below 11.1 nm) to both BRI1 and BAK1 in vivo. In the radius of 11.1 nm, the three proteins can form trimers and/or be arranged as intermediate complexes (Figure 6). Application of 10 nM BL to the cells did not change the FRET characteristics in the spatial RLP44-mTRQ2/BRI1-mVEN/BAK1-mRFP arrangement in the PM. This indicates that the RLP44/BAK1/BRI1 nanodomain is pre-formed in the PM in the absence of BL and none of the proteins appears to leave the nanodomain upon binding of BL to BRI1 to a significant amount.

According to our intensity-based FRET and FRET-FLIM data, FLS2 interacts with neither BRI1 nor RLP44, which is in agreement with recent BiFC studies [30]. Remarkably, the spatial distance between FLS2 and the RLP44-BAK1, RLP44-BRI1 and the RLP44-BRI1-BAK1 heteromers must be larger than 11.1 nm. This indicates that FLS2 is localized in nanodomains that are spatially distinct from the RLP44-consisting ones. Taking into account the around 20-fold better nanometer accuracy of FRET, our approach decisively proves previous results, indicated by variable angle epifluorescence microscopy (VAEM) which has a physical resolution limit of roughly 250 to 300 nm [29,31].

In summary, using the example of three selected chromophores, we show that three-fluorophore intensity-based FRET and three-fluorophore FRET-FLIM are suitable techniques for analyzing the interaction and relative proximity of different proteins in the PM of living plant cells. Furthermore, by applying the presented calculations and experimental approaches, it is possible to infer spatially distinct complexes and/or nanodomains in the PM that are different in their protein composition.

Although the two approaches are complementary, we strongly recommend the FRET-FLIM approach rather than intensity-based, spectral FRET for ternary protein interaction and proximity analyses, as the latter is particularly vulnerable to changes in the donor-to-acceptor ratio and other technical restrictions.

## 4. Material and Methods

### 4.1. Plasmid Construction

The cDNA sequence of the gene of interest without stop codon was brought into pDONR221-P3P2 (donor) or pDONR221-P1P4 (first acceptor) or pENTR™/D-TOPO^®^ (second acceptor) as described by guidelines in the Gateway manual (Life Technologies, Schwerte, Germany) with primers listed in Appendix A. The coding sequence of BRI1 and BAK1 was brought in the pENTR-D-TOPO previously [7,8]. For the generation of BRI1-HA, primer previously published were used [9] to bring CDS of BRI1 in pDONR207 and an LR with pGWB14 was performed. The LR into pB7RWG2 (RFP) [44] and the 2in1 FRET vector pFRETtv-2in1-CC [15] was performed as described previously [15,45].

### 4.2. Localization and FRET-FLIM Studies

Transformation of *N. benthamiana* was performed as described by [15,46], omitting the washing step with sterile water. For transformations with multiple constructs, an OD_600_ of 0.1 was set and mixed 1:1:1 with silencing inhibitor p19. Plants were watered and left at ambient conditions (24 °C) with the lid on top and imaged two days past transformation with an SP8 confocal laser scanning microscope (CLSM) (Leica Microsystems GmbH, Wetzlar, Germany) with LAS AF and SymPhoTime software (Picoquant GmbH, Berlin, Germany) using a 63×/1.20 water immersion objective [46,47]. Data were derived from measurements of the lower epidermis, avoiding guard cells and stomata, with at least two biological replicates, comprising in average 20 data points and 11 data points for mTRQ2—mRFP controls. Localization and quantification were performed with a minimum 3-fold line average for mTRQ2, mVEN and mRFP with the Argon laser set to 2% and excitations of 458 nm 40%, 514 nm 20% and 594 nm or 561 nm 30% and emission detection with 465–505 nm 400% on HyD, 525–565 nm 400% on SMD HyD and 605–650 nm 300% on SMD HyD, respectively.

FLIM measurements were performed with a 440 nm pulsed laser (LDH-P-C-470, Picoquant GmbH, Berlin, Germany) with 40 MHz repetition rate at a reduced scanning speed, yielding, with an image resolution of 256 × 256, a pixel dwell time of ~20 µs. The maximal count rate was set to ~2000 cps. Measurements were stopped when the brightest pixel had a photon count of 500. Only measurements with an even intensity distribution at the PM were included. The corresponding emission was detected with HyD SMD from 455 nm to 505 nm by time-correlated single-photon counting using a PicoHarp 300 module or a TimeHarp 260 module (PicoQuant GmbH, Berlin, Germany). The calculation of FLTs was performed by iterative reconvolution, i.e., the instrument response function was convolved with an exponential test functions to minimize the error with regard to the original TCSPC histograms in an iterative process. While the donor-only samples were fitted with mono-exponential decay functions, the energy transfer to fluorophores in the other samples resulted in additional decay rates. These histograms necessitated biexponential fitting functions, from which the fluorescent lifetime was derived by intensity weighted averaging. For the fastFLIM measurements, the maximal count rate was increased.

### 4.3. Acquisition of λ-Stacks (Spectra)

Expression of relevant fluorophores were checked via fluorescence level prior λ-stack acquisition. For λ-stacks, both sequential excitation and simultaneous excitation was used as mentioned in Results and an average of at least 6 ROIs of the PM with different expression levels of at least two biological replicates. At the Leica SP8 microscope, excitation at 458 nm 80% was used with SMD HyD ~250%, measuring 460–625 nm with Δ7.5 nm and 256 × 256 px resolution, a pixel dwell time of ~20 µs and three-fold line accumulation. At the Zeiss LSM880 (Carl Zeiss AG, Oberkochen, Germany) excitation at 485 nm with 30%, NF458, 800 V of airy-scan detectors were used, measuring 460–650  nm or 560–650 nm with Δ4.5 nm and 256 × 256 px resolution, speed 2, digital gain set to 2, pinhole set to 14.25 airy units and a three-fold line accumulation. If over-all expression was very high, then for all samples measured that day line average was taken instead. For additional information see Appendix A.

### 4.4. Protein Structures and Sizes

The intracellular domain of BRI1 (5LPW) and BAK1 (3TL8) as well as the fluorophore barrels were exported from the Protein Data Base (PDB). As viewer, Jmol: an open-source Java viewer for chemical structures in 3D [48] with solvent accessible depiction was chosen and protein colors were changed. The secondary structure of linkers and the intracellular RLP44 domain was predicted with PEP-FOLD 3.5 [26], de-novo prediction, with standard settings and always model no1 (of 10) was used. Amino acid sequences were HPTFLYKVGQLLGTS for the donor-linker, NPAFLYKVVSRLGTS for the acceptor-linker, KGGRADPAFLYKVVIS for the second acceptor linker and CLWLRITEKKIVEEEGKISQSMPDY for RLP44_Cyto_. The size of intracellular domains was calculated from the known distance of the alpha-barrel secondary structure, which is Δ5.4 Å from turn-to-turn and 4 Å inner diameter.

### 4.5. Statistics

With one exception, each measurement was performed in at least three biological replicates. Each biological replica in turn included at least 3 individual recordings. In the exception, 2 biological replicates were performed, each containing 3 individual recordings. Images and plots were generated with Microsoft Excel v1809, SAS JMP 14 or MATLAB [49], also using these programs for calculation of average, standard error (SE) and standard deviation (SD). To test for homogeneity of variance, Levene’s test (*p* < 0.05) was employed and statistical significance for non-parametric distributions was calculated by a two-tailed, all-pair Kruskal-Wallis test followed by a Steel-Dwass *post hoc* correction using SAS JMP version 14.0.0 [50]. For small sample numbers in Figure 3d the 2-sample *t*-test was chosen [51].

## Figures and Tables

**Figure 1 plants-11-02630-f001:**
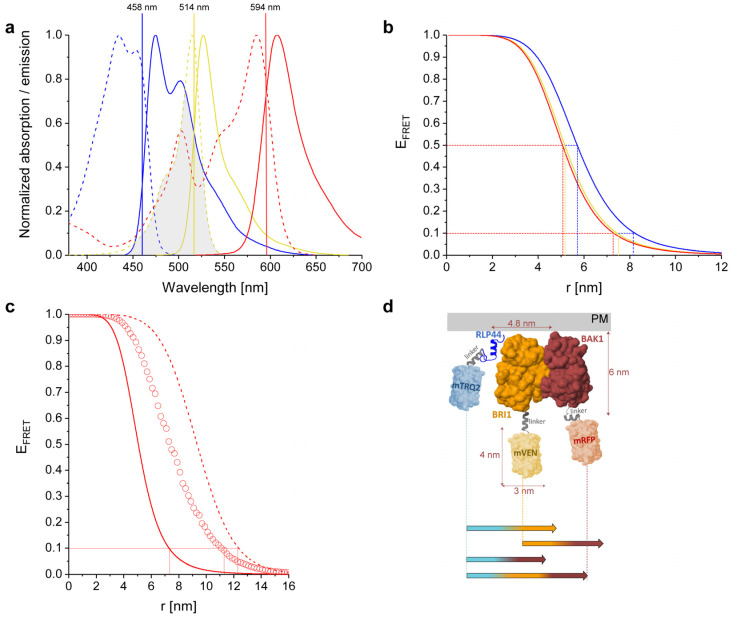
Spectroscopic and FRET properties of the used fluorophores and dimensions of the analyzed RLP44-mTRQ2, BAK1-mVEN and BRI1-mRFP fusion proteins. (**a**) Normalized absorption (dotted lines) and emission (solid lines) spectra of mTRQ2 (blue), mVEN (yellow) and mRFP (red). mTRQ2 is donor to both mVEN as acceptor 1 as well as mRFP as acceptor 2; mVEN is also donor to mRFP. The shaded area corresponds to the spectral overlap between donor emission and acceptor absorption, shown here exemplary for the mTRQ2/mVEN FRET pair. The laser lines for excitation of mTRQ2 (blue), mVEN (yellow) and mRFP (red) are marked as vertical lines at their respective wavelength positions. (**b**) FRET efficiencies (E_FRET_) for the distances (r) between donor and acceptor fluorophores for mTRQ2-mVEN (blue), mVEN-mRFP (yellow) and mTRQ2-mRFP (red). The Förster distances R_0_ at E_FRET_ = 50% as well as the distances that correspond to E_FRET_ = 10% (r_10%_) for each pair are marked with dashed lines in the respective color. (**c**) E_FRET_ in dependence of distance r between mTRQ2 and mRFP without (solid line) and with intermediate mVEN located either equidistantly (dashed line) or variably (circles) between mTRQ2 and mRFP. For the latter, mVEN was placed at 1000 random positions for each mTRQ2/mRFP distance, averaging over the resulting E_FRET_ values. The r_10%_ distances are marked with red lines. (**d**) Composite image of the cytoplasmic domains of RLP44 (blue), BRI1 (orange) and BAK1 (brown) fused to the 15 amino acid-long Gateway^®^-linker to either mTRQ2 (light blue), mVEN (light yellow) or mRFP (light red). The structures of the cytoplasmic domains of BAK1, BRI1 and the fluorophores are shown as solvent-accessible surface models, while the model structures of the cytoplasmic domain of RLP44 and the three Gateway^®^-linkers (grey) were predicted with PEP-FOLD3 and depicted as cartoons. The orientation of the kinase domains of BRI1 and BAK1 to each other was designed according to a molecular docking analysis in orientation to the PM [20]. The colored arrows below the structures show the r_10%_ distances for all FRET pairs and the three-chromophore FRET cascade with mVEN inserted at random positions, calculated for a donor to acceptor complex ratio of 1:1. The precise values for b, c and d are listed in Table 1.

**Figure 2 plants-11-02630-f002:**
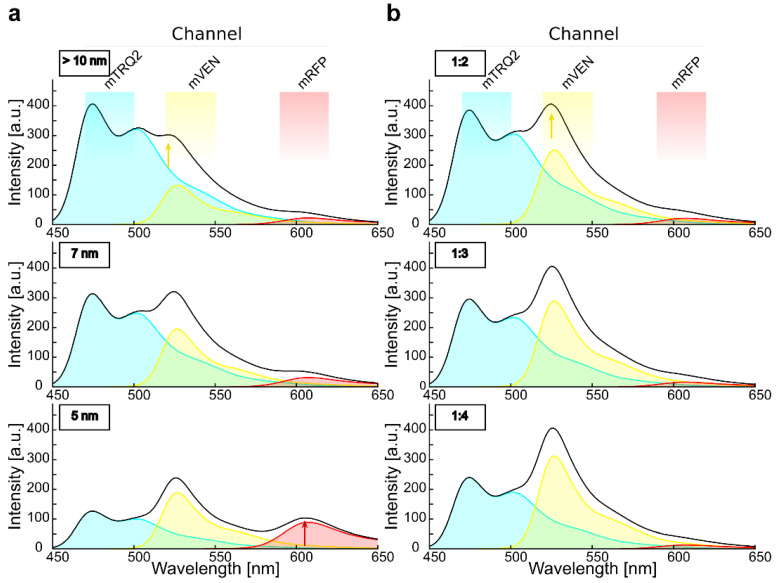
Simulation of emission spectra reveals the crucial impact of relative fluorophore quantities for intensity-based spectral FRET in a three-chromophore arrangement. (**a**) Simulation of the spectra that result from excitation of mTRQ2 with 458 nm for a mTRQ2/mVEN/mRFP stoichiometry of 1:1:1 and in a linear arrangement while placing mVEN at an equal distance from the other two fluorophores. The emission spectra of mTRQ (blue area), mVEN (yellow area) and mRFP (red area) and their combined spectrum (black line) are calculated. The respective emission detector channels are depicted above the plots. With inter-fluorophore distances larger than 10 nm, FRET is negligible (**top**), but cross-excitation of mVEN (yellow arrow) is apparent. For inter-fluorophore distances of 7 nm (**middle**) and 5 nm (**bottom**), FRET lowers the emission intensity of mTRQ2, while the intensity peaks increases around the emission of mVEN at 525 nm and of mRFP at about 610 nm (red arrow) increase. (**b**) Spectra simulation as in (**a**), but without consideration of FRET, calculated for different mTRQ2/mVEN fluorophore ratios. With rising mTRQ2/mVEN ratios of 1:2 (**top**), 1:3 (**middle**) and 1:4 (**bottom**), the intensity peak of mVEN (yellow arrow) increases. Thus, a mTRQ2/mVEN ratio of 1:2 ((**b**), top) results in a similar spectral shape than with equimolar fluorophores subjected to FRET at a distance of 7 nm ((**a**), middle).

**Figure 3 plants-11-02630-f003:**
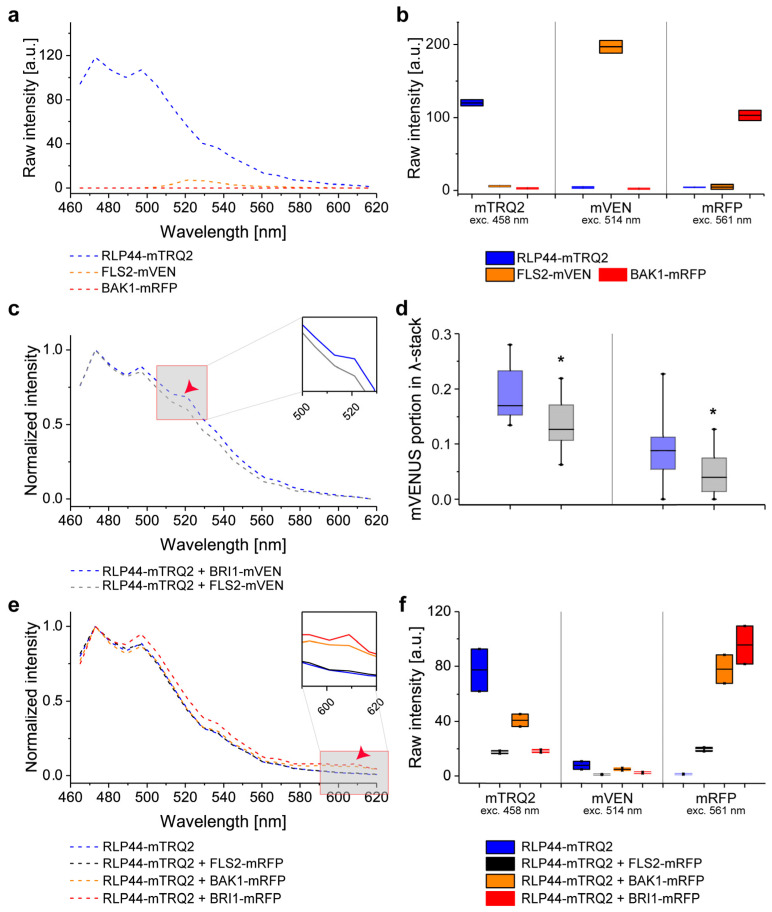
Interaction of RLP44 with BAK1 and BRI1 is detectable in the plasma membrane of *N. benthamiana* epidermal leaf cells by intensity-based spectral FRET. (**a**) Wavelength-dependent intensity of fluorescence emission after irradiation of the cells with 458 nm light for RLP44-mTRQ2 (blue), FLS2-mVEN (orange) and BAK1-mRFP (red). (**b**) Emission intensity after irradiation with light of different wavelengths in the mTRQ2 (left, 458 nm), mVEN (middle, 514 nm) and mRFP channels (right, 561 nm) from 8-bit images for RLP44-mTRQ2 (blue), FLS2-mVEN (orange) and BAK1-mRFP (red) after correction for spectral bleed-through. (**c**) Wavelength-dependent normalized fluorescence emission for RLP44-mTRQ2 co-expressed with BRI1-mVEN (blue) or with FLS2-mVEN (grey). For both spectra, the average donor-to-acceptor ratio was around 1:4 (0.26). The FRET-relevant wavelength area is as an inset and the occurrence of FRET from RLP44-mTRQ2 to BRI1-mVEN is indicated by a red arrow head. The data are presented as the mean (n ≥ 13). (**d**) Protein-dependent, relative mVEN emission signal after irradiation of the cells with light of 458 nm light. Fusion protein ratios of about 1:4 (left) and 1:10 (right) of the RLP44-mTRQ2/BRI1-mVEN pair (blue) and RLP44-TRQ2/FLS2-mVEN pair (grey) are shown. Significant changes [*n* = 7 (left), *n* = 12 (right)] in a two-sided 2-sample *t*-test with *p* < 0.05 are indicated by asterisks. (**e**) Wavelength-dependent normalized fluorescence emission after irradiation of the cells with 458 nm light for RLP44-mTRQ2 alone (blue), RLP44-mTRQ/FLS2-mRFP (black), RLP44-mTRQ/BAK1-mRFP (orange) and RLP44-mTRQ/BRI1-mRFP (red). The FRET-relevant wavelength area is highlighted in an enlarged section and the occurrence of FRET fromRLP44-mTRQ2 to BAK1-mRFP or BRI1-mRFP is indicated by a red arrow head. (**f**) Emission intensity after irradiation of the cells with light of different wavelength in the mTRQ2 (left), mVEN (middle) and mRFP channel (right) from 8-bit images for of RLP44-mTRQ2 (blue), RLP44-mTRQ/FLS2-mRFP (black), RLP44-mTRQ/BAK1-mRFP (orange) and RLP44-mTRQ/BRI1-mRFP (red). For further details see (**b**). For the statistical evaluation see Section 4. Boxplots in (**b**) and (**f**) represent the measured data with the average as a solid black line and the box limits are the mean +/− standard deviation. The boxplot in (**d**) represents all data with the median as a solid black line within the box that is restricted by the first quartile (25%; lower end) and the third quartile (75%; upper end). Whiskers show the minimum and maximum of the measurements, respectively.

**Figure 4 plants-11-02630-f004:**
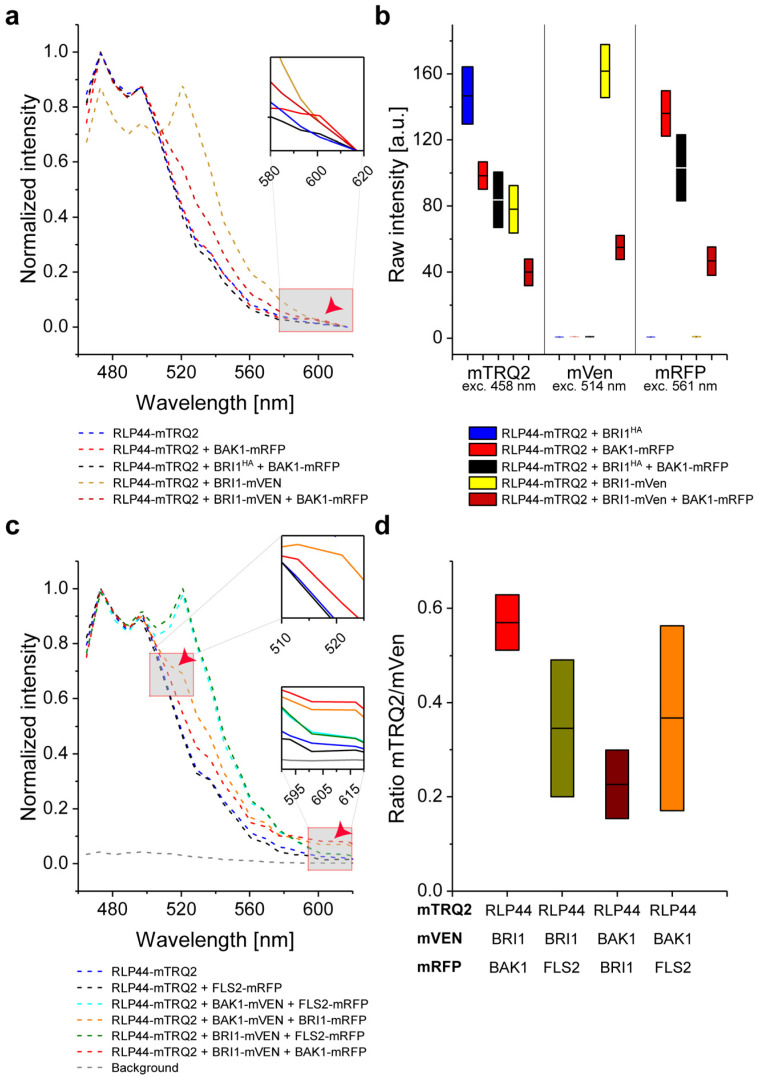
The formation of a ternary complex consisting of RLP44, BAK1 and BRI1, but not FLS2, is detectable in the plasma membrane of *N. benthamiana* epidermal leaf cells by intensity-based spectral FRET. (**a**) Wavelength-dependent normalized fluorescence emission after irradiation of the cells with 458 nm light for RLP44-mTRQ2 alone (blue) or co-expressed with either BAK1-mRFP (light red) or HA-tagged BRI1 (BRI1^HA^) and BAK1-mRFP (black) or BRI1-mVEN (bronze) or BRI1-mVEN and BAK1-mRFP (dark red). The FRET-relevant wavelength area is highlighted in the inset image. The occurrence of FRET from mTRQ2 to mRFP is indicated by a red arrowhead. (**b**) Emission intensity after irradiation of the cells with light of different wavelengths in the mTRQ2 (left, 458 nm), mVEN (middle, 514 nm) and mRFP channels (right, 561 nm) from 8-bit images for RLP44-mTRQ2 co-expressed with either BRI1^HA^ (blue), BAK1-mRFP (light red), BRI1^HA^ and BAK1-mRFP (black), BRI1-mVEN (yellow) or BRI1-mVEN and BAK1-mRFP (dark red). (**c**) Wavelength-dependent normalized fluorescence emission after irradiation of the cells with 458 nm light for the expression of RLP44-mTRQ alone (blue) or co-expression with either BRI1-mVEN and BAK1-mRFP (red) or BAK1-mVEN and BRI1-mRFP (orange) or BRI1-mVEN and FLS2-mRFP (green) or BAK1-mVEN and FLS2-mRFP (cyan) or with FLS2-mRFP (black). The FRET-relevant wavelength areas are highlighted in enlarged sections. The occurrence of FRET from mTRQ2 to mRFP around 610 nm and the FRET-caused decrease of the mVEN signal around 525 nm are indicated by red arrowheads. The cell´s background emission of the cells is shown as dashed grey line. (**d**) Fluorescence emission ratio after irradiation of the cells with light of different wavelength in the mTRQ2 channel (**left**), mVEN channel (**middle**) and mRFP channel (**right**) from 8-bit images for RLP44-mTRQ2 co-expressed with BRI1^HA^ and BAK1-mRFP (black), BRI1-mVEN (yellow) and BRI1-mVEN and BAK1-mRFP (dark red). See (**b**) for further details. The boxplots in (**b**) and (**d**) represent all data with the average (here equivalent to the median) as a solid black line within the box that is restricted by +/− standard deviation. For the statistical evaluation see Section 4.

**Figure 5 plants-11-02630-f005:**
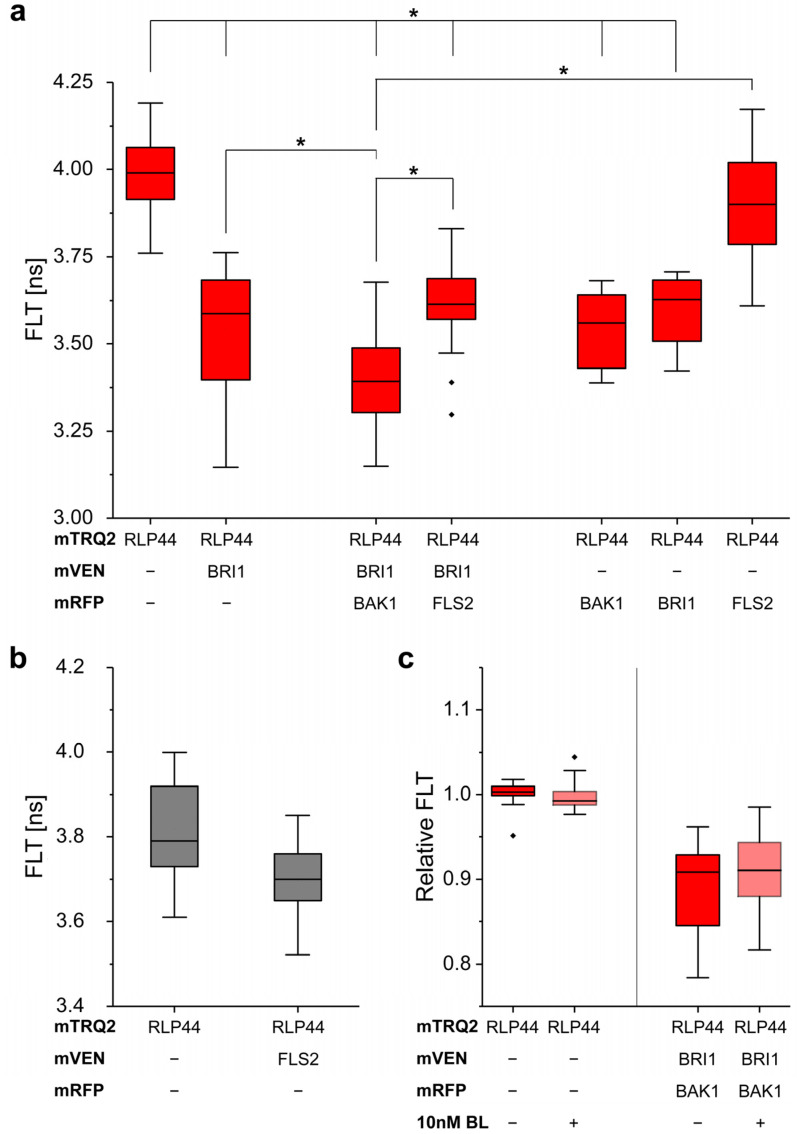
FRET-FLIM enables quantitative three-fluorophore protein-protein interaction and proximity analyses in *N. benthamiana* leaf cells. (**a**) Fluorescence lifetime (FLT) of RLP44-mTRQ2 after pulsed excitation of the cells with 440 nm light in the presence of the indicated mVEN and mRFP fusion proteins. (**b**) FLT of RLP44-mTRQ2 alone or in the presence of FLS2-mVEN. Data presentation as in (**a**). (**c**), Relative FLT of RLP44-mTRQ2 alone or in the presence of the indicated fusion proteins without (dark red) and with application of 10 nM brassinolide (BL) (light red). The average FLT of RLP44-mTRQ2 in the absence of BL was set to 1. The boxplots represent all data with the median as a solid black line within the box that is restricted by the first quartile (25%; lower end) and the third quartile (75%; upper end). Whiskers show the minimum and maximum value of the data, respectively, that are not defined as outlier (1.5 times interquartile range). Outliers are indicated as black diamonds. Statistical evaluations were performed by ANOVA followed by Tukey-Kramer HSD *post hoc* test. The black asterisk indicate statistically significant differences (*p* ≤ 0.005).

**Figure 6 plants-11-02630-f006:**
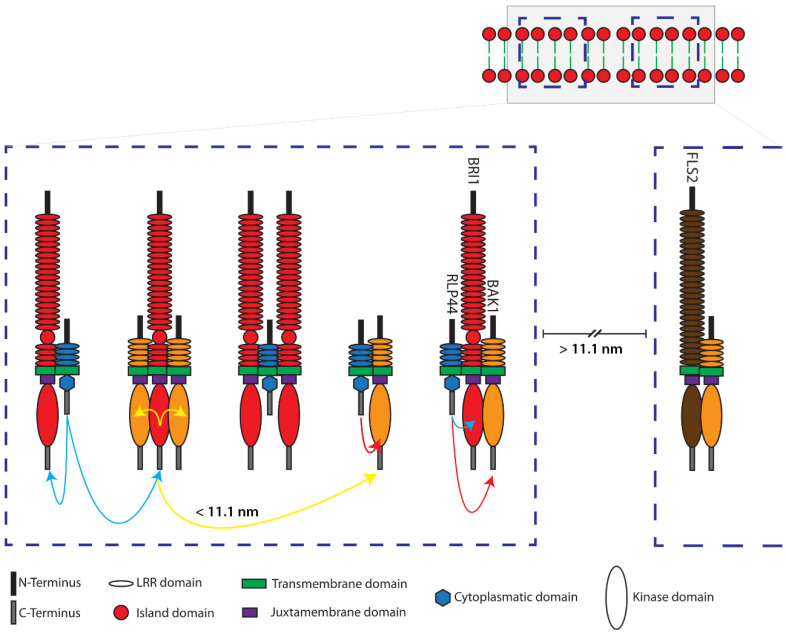
Model of the ternary interactions and spatial distances of RLP44, BRI1, BAK1 and FLS2 in defined nanodomains located in the plasma membrane of plant cells, as determined by three-fluorophore, intensity-based spectral FRET or FRET-FLIM. Shown are two distinct schematic nanodomains (blue dashed squares in the top corner) in the plasma membrane. The left nanodomain in the enlargement consists of RLP44-mTRQ2 (blue), BRI1-mVEN (red) and BAK1-RFP (orange)—without the fluorophores and not in scale. The formation of different complexes is demonstrated by FRET between mTRQ2 and mVEN by blue, mVEN and mRFP by yellow and mTRQ2 and mRFP by red arrows. FRET is also possible between the fusion proteins of differently composed complexes, if they are in a distance of ≤11.1 nm (representative yellow arrow). Due to the absence of FRET, for instance, the BAK1-mVEN/FLS2-mRFP (brown) complex must be at least 11.1 nm separated from the nanodomain, that contains the RLP44 related complexes, which is very likely part of an independent nanodomain (right side in the enlargement).

**Table 1 plants-11-02630-t001:** FRET combinations (1st column), their Förster distances (R_0_) (2nd column) and fluorophore distances that correspond to 10% FRET efficiency (r_10%_), calculated for a linear arrangement with one acceptor (3rd column) in range of each corresponding donor (see also Appendix A).

FRET Combinations	R_0_ [nm]	r_10%_ [nm], D:A = 1:1
mTRQ2-mVEN	5.7	8.2
mVEN-mRFP	5.2	7.4
mTRQ2-mRFP	5.1	7.3
mTRQ2-mVEN-mRFP (middle position)	-	12.4
mTRQ2-mVEN-mRFP (random position)	-	11.1

## Data Availability

All data supporting the findings of the study are present in the main text and/or the Appendix A. Source data are provided with this paper. All MATLAB code for the calculation of the values in Table 1 and the spectral unmixing is available at https://github.com/svenzok/3F-FRET (accessed on 23 September 2022).

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
