# Peer review of "Three-Fluorophore FRET Enables the Analysis of Ternary Protein Association in Living Plant Cells"

_plants, 2022, doi:10.3390/plants11192630_

Round 1
Reviewer 1 Report
In this study, the authors showed that three-fluorophore FRET enables the analysis of ternary protein association in living plant cells. To be specific, they used intensity-based FRET and FRET-FLIM to prove that RLP44 is located in close proximity to both BRI1 and BAK1 in vivo. In addition, they showed that FRET-FLIM is better than intensity-based FRET in probing ternary protein association in living plant cells. The experiments were well-designed and executed, the results are soundness and supports their conclusion.
Some minor issues should be addressed before publication:
1. I suggest show one confocal image with ROI in the maintext to indicate where and how the spectra was taken/measured.
2. Page 1, second paragraph in Introduction, "(review as reference)" should be removed?
3. Page 2, second paragraph, "...., we at first had to establish...": before introducing Three-fluorophore FRET, here "we had to" sounds no logical. I suggest delete "had to".
4. Page 2, third paragraph, "In addition, we propose FRET-FLIM to be always the better choice for the analysis of ternary protein complex formation and relative proximity estimates in plant cells." This sentence confuses the readers because it reads like FRET-FLIM is the best among all approaches. I suggest change to "In addition, we propose FRET-FLIM to be always better than intensity-based FRET for the analysis of ternary protein complex formation and relative proximity estimates in plant cells."
5. Page 15: "The boxplots represent all data ...The black asterisk indicate statistically significant differences (p≤ =0.005)." Is this paragraph part of the lengend of Figure 5? If yes, the format need to be changed.
6. Page 19: "Funding: Funding:" : repeated wording here.
Author Response
Dear Reviewer,
Thank you for your helpful notes and suggestions. Please find our response below.
Sincerely yours,
Klaus Harter
Ad 1. : If we incorporate the ROIs and data analysis here, it interferes with the flow of the text. We have now inserted a sentence into the end of the first paragraph of chapter 2.5 (page 6) that explicitly refers to the selection of ROIs, data collection and data analysis in figure 6S and Supplementary Materials.
Ad 2., page: "(review as reference)" was deleted
Ad 3., page 2: The sentence was changed to "we first wanted to establish..."
Ad 4., page 2: We changed the sentence according to the reviewer´s suggestion.
Ad 5., page 15: The format of the figure legend was adjusted
Ad 6., page 19: The doubled word "funding" was removed.
Reviewer 2 Report
This manuscript describes a state-of-the-art design of a three-fluorophore FRET system and its application for probing the formation of the ternary complex of receptor-like protein 44 (RLP44) with hormone-binding receptor BRI1 and its associated kinase (BAK1) in the membranes of plant cells. Despite several potential bottlenecks and the high complexity of the three-fluorophore FRET formalism, the authors succeeded in implementing this approach in both the fluorescence intensity and FLIM setup and demonstrated the formation of the RLP44/BAK1/BRI1 complex. This study opens a path for further application of three-fluorophore FRET in the studies of protein-protein interactions in membranes. The manuscript is well-written and relatively easy to understand, despite the high complexity of the matter. I have only a few minor concerns:
1. All abbreviations used in the manuscript (RLP44, BRI1, BAK1, etc.) must be defined in the main text (not in the abstract) at their first occurrence.
2. Page 1, line 8 from the bottom: “(review as reference)” – It looks like the authors meant to insert a reference to some review here but forgot to do so.
3. Section 2.1 – this section does not sound like a part of the Results because none results are described here. It is rather a part of the Introduction. I would suggest placing this paragraph before the final paragraph of the Introduction (right before “Based on theoretical calculations…”)
4. Paragraph starting at line 17 on page 4 and equation (12) and surrounding text in the Supplement: I believe the discussion of the “antenna” effect is irrelevant here. According to my understanding, the authors always imply specific interactions in the RLP44-BAK1-BRI1 triad and assume 1:1:1 stoichiometry of the complex. However, the antenna effect formalism applies only to the complexes of one donor with multiple similar and equidistant acceptors. It is unclear how the crowding in the membrane may promote the formation of complexes of this type if the interactions are 1:1:1 and specific. Otherwise, if the authors mean unspecific, crowding-promoted interactions, the scientific value of this study becomes questionable. Therefore, I would suggest dropping all discussions of the antenna effect from the paper. If the authors decide to keep it, they may need to add a more detailed discussion on the stoichiometry and specificity of the interactions.
Author Response
Dear Reviewer,
Thank you for your helpful hints and comments. We have tried to take them into account in the text - hopefully to your satisfaction.
Sincerely yours,
Klaus Harter
Ad 1.: We defined the abbreviations also in the main text.
Ad 2.: "(review as reference)" was deleted
Ad 3.: We would like to keep this paragraph in the Results. The reason is that the selection of fluorophores was preceded by an extensive screen, which resulted in the use of the specified fluorophores. We have reworded the opening sentence of this chapter with this mind.
Ad 4.: The reviewer raises an important point here that we have not made sufficiently clear. In fact, our goal was only to draw general attention to the intricacies of complex fluorophore arrangements. As the reviewer had noted, the antenna effect is formulated only for specific FRET configurations between one and multiple acceptors. However, our model is based on the assumption of specific 1:1:1 interactions between three different fluorophores and does not involve unspecific interactions due to crowding. Our formulation unintentionally suggested that the antennae effect and the also mentioned FRET surplus were integral parts of our model. We hereby follow the reviewer's suggestion to remove this misleading discussion. Ultimately, this paragraph is only meant to emphasize that the biological reality is presumed more complex, but that we can already use our simplified model to explain how we can detect FRET even over longer distances. Accordingly, we have also removed the approximated calculations based on these effects from the main text and now refer only to the theoretical results of 1:1 donor-acceptor ratios, lowering our calculated maximum dynamic FRET range and the derived minimum spatial complex distances from 13.1 nm to a more conservative 11.1 nm. Since this is still a significant improvement over the FRET ranges in classical two-fluorophore experiments, our discussions and conclusions remain unaffected by this numerical change.